# Towards Practical Second Order Optimization for Deep Learning

## Abstract

Optimization in machine learning, both theoretical and applied, is presently dominated by first-order gradient methods such as stochastic gradient descent. Second-order optimization methods, that involve second derivatives and/or second order statistics of the data, are far less prevalent despite strong theoretical properties, due to their prohibitive computation, memory and communication costs. In an attempt to bridge this gap between theoretical and practical optimization, we present a scalable implementation of a second-order preconditioned method (concretely, a variant of full-matrix Adagrad), that along with several critical algorithmic and numerical improvements, provides significant convergence and wall-clock time improvements compared to conventional first-order methods on state-of-the-art deep models. Our novel design effectively utilizes the prevalent heterogeneous hardware architecture for training deep models, consisting of a multicore CPU coupled with multiple accelerator units. We demonstrate superior performance compared to state-of-the-art on very large learning tasks such as machine translation with Transformers, language modeling with BERT, click-through rate prediction on Criteo, and image classification on ImageNet with ResNet-50.

## 1 Introduction

Second order methods are among the most powerful algorithms in mathematical optimization. Algorithms in this family often use a preconditioning matrix to transform the gradient before applying each step. Classically, the preconditioner is the matrix of second-order derivatives (i.e., the Hessian) in the context of exact deterministic optimization (e.g., Fletcher, 2013; Lewis & Overton, 2013; Nocedal, 1980). While second-order methods often have significantly better convergence properties than first-order methods, the size of typical problems prohibits their use in practice, as they require quadratic storage and cubic computation time for each gradient update. Approximate algorithms such as quasi-Newton methods are aimed at significantly reducing these requirements; nonetheless, they still impose non-trivial memory costs equivalent to storing several copies of the model (and often quadratic computation, as in the popular two-loop recursion (Nocedal, 1980)), which severely limits their use at the immense scale of present-day deep learning.

Arguably, one of the greatest challenges of modern optimization is to bridge this gap between theoretical and practical optimization towards making second-order methods feasible to implement and deploy at immense scale. Besides the compelling scientific and mathematical developments it may stimulate, this challenge has also a clear real-world significance: recent practice of training deep learning models suggests that the utility of common first-order methods is quickly reaching a plateau, in large part because their time-per-step is already negligible (compared to other parts of the computation) and cannot be optimized further; thus, the only way to obtain faster training performance is by drastically reducing the number of update steps. To this end, utilizing second-order methods seem a very natural and promising approach.

In this paper we attempt to narrow the gap between theory and practice of second-order methods, focusing on second-order *adaptive* methods for stochastic optimization. These methods can be thought of as full-matrix analogues of common adaptive algorithms such as AdaGrad (Duchi et al., 2011; McMahan & Streeter, 2010) and Adam (Kingma & Ba, 2014): they precondition each gradient with a second moment matrix, akin to a covariance matrix, that accumulates the outer products of the stochastic gradients. Full-matrix versions are potentially more powerful than first-order methods as they can exploit statistical correlations between (gradients of) different parameters; geometrically,

they can scale and rotate gradients whereas first order methods only scale gradients. However they suffer from similar prohibitive runtime and memory costs as Hessian-based methods.

Recent developments in the space of second-order methods, on which we focus on in this paper, include the K-FAC (Heskes, 2000; Martens & Grosse, 2015) and Shampoo (Gupta et al., 2018) algorithms that exploit the structure of deep networks (and more generally, models described by a collection of tensors) for mitigating the space and runtime costs of full-matrix second-order algorithms. These methods approximate each preconditioning matrix using a factored representation that stems from the network structure. However, in very large applications, such algorithms are still impractical due to a number of numerical and infrastructural pitfalls and are difficult to parallelize.

*Contributions.* We provide solutions to practical concerns and challenges that arise in implementing and using second-order methods at large scale. Our focus will be on the Shampoo algorithm, but most of the challenges we address are relevant to the implementation of many other second-order methods. These include:

- We design and implement an pipelined version of the optimization algorithm, critically exploiting the heterogeneity and computing power of CPU-Accelerator coupled architectures;
- We extend Shampoo in a number of ways so as to make it applicable to a larger range of deep architectures; in particular, the extensions allow Shampoo to be used for training very large layers such as embedding layers ubiquitous in language and translation models;
- We replace expensive spectral decompositions (e.g., SVD) used for manipulating preconditioners with an efficient and numerically-stable iterative method for computing roots of PSD matrices;
- We describe practical challenges and limitations we faced in our design, which we argue could be useful for the design considerations of next-generation accelerator hardware architectures.

Our distributed implementation demonstrates significant improvements in performance, both in terms of number of steps, and often in actual wall-clock time, on some extremely large deep learning tasks:

- *Machine translation*: we train Transformer models (Vaswani et al., 2017) on the WMT'14 English to French translation task (Bojar et al., 2014) in *half as many steps* compared to state-of-the-art (well tuned Adam), resulting with *up to 45% reduction in wall-time*.
- *Language modeling*: we trained BERT (Devlin et al., 2018) in *16% fewer steps* and achieve *higher masked-LM accuracy* compared to state-of-the-art optimizer (You et al., 2019) at 32K batch size; *overall wall-time decreased by 4% from 3.8 to 3.65 hours*. (For this task, our system has not yet been tuned for performance; we discuss several possible optimizations below.)
- *Click-Through Rate (CTR) prediction*: we trained the DLRM model (Naumov et al., 2019) on the terabyte Criteo dataset (Criteo Labs, 2015) at 64K batch size in *half as many steps* as the current state-of-the-art optimizer, with a *wall-time reduction of 37.5%*. We achieve a new state-of-the-art performance of 80.56% AUC ($\approx 0.3\%$ improvement) on this task. (An improvement of 0.1% is considered significant; see Rong et al., 2020; Wang et al., 2017.)
- *Image classification*: we achieve MLPerf target accuracy of 75.9% (Mattson et al., 2019) at 32K batch size on the standard ResNet-50 ImageNet benchmark in *10% fewer steps* than previous state-of-the-art. Here we do not see wall-time gains, mainly because the problem is too small (only few thousand steps for convergence which does not allow for amortization of costs). However, we expect that one would be able to better exploit parallelism via improved software and hardware support.

We note that one of our main points in this work was to demonstrate wall-time speedups with second-order methods implemented on a *real-world distributed setup* being used to train state-of-the-art deep models. In our view, this is important for influencing future hardware accelerator design and runtime software. Indeed, first-order methods have received huge investments in tuning, implementation, platform support and tailored accelerator hardware over the last decade; we believe there are numerous opportunities to improve the per-step time performance of preconditioned methods as well. For example, our results provide a concrete justification for incorporating 64bit accumulation units in hardware for distributed training, adding larger on-chip memory, better model parallelism and tighter coupling between accelerators and CPUs, which would make second order methods feasible across more domains and models.

*Related work.* Classic techniques for addressing the high storage and computation costs of second-order methods mostly belong to the quasi-Newton or the trust-region families of algorithms (Conn et al., 2000; Nocedal & Wright, 2006). Traditionally, these methods need nearly-accurate gradients in

order to construct useful quadratic approximations and implement reliable line searches, rendering them as suitable for training with very large batch sizes, and resulting in expensive iterations that make the overall algorithm slow compared with stochastic first-order methods (see, e.g., Bollapragada et al., 2018 for a recent account). Hence, our focus in this paper is on adaptive second-order methods which are directly applicable in a stochastic setting. That said, our effort could be relevant to quasi-Newton and trust-region methods as well: e.g., each iteration of typical trust-region methods amounts to solving a certain generalized eigenvalue problem, which presents numerical difficulties of similar nature to those encountered in matrix root/inverse computations, being addressed here.

Various approximations to the preconditioning matrix have been proposed in the recent literature (e.g., Gonen & Shalev-Shwartz, 2015; Erdogdu & Montanari, 2015; Agarwal et al., 2016; Xu et al., 2016; Pilanci & Wainwright, 2017). However, so far the only prevalent and pragmatic approximation is the diagonal approximation. Some recent approaches for approximating a full-matrix preconditioner are K-FAC (Martens & Grosse, 2015), Shampoo (Gupta et al., 2018) and GGT (Agarwal et al., 2018). K-FAC uses a factored approximation of the Fisher-information matrix as a preconditioner. While our focus in this paper is on Shampoo, we believe that many of the techniques presented here could also be applied to make K-FAC practical in large scale (see Appendix C). GGT uses a clever trick to compute a low-rank approximation to the AdaGrad preconditioner. However, GGT maintains several hundred copies of the gradient in memory, which is too expensive even for mid-sized models.

Ba et al. (2017) took a first important step at experimenting with distributed K-FAC for training deep models, using a single machine with 8 GPUs to simulate a distributed environment for training. In contrast, a main thrust of our work is to demonstrate wall-time speedups with second-order methods on a real-world distributed setup used for training state-of-the-art deep models, that call for design considerations crucially different than in (Ba et al., 2017). More recently, Osawa et al. (2019) scaled up K-FAC for training convolutional networks, but fell short of reaching the accuracy of first order methods, despite making changes to data augmentation and model architecture.

## 2  PRELIMINARIES

*Adaptive preconditioning methods.* First order methods iteratively update the parameters solely based on gradient information: $w_{t+1} = w_t - \eta_t \bar{g}_t$ where $w_t$ and $\bar{g}_t$ are (column) vectors in $\mathbb{R}^d$. Here $\bar{g}_t$ denotes a linear combination of the current and past gradients $g_1, \ldots, g_t$, where different algorithms use different combinations. Preconditioned methods take the form $w_{t+1} = w_t - P_t \bar{g}_t$ where $P_t$ is an $d \times d$ matrix. Whereas in Newton-type methods this matrix is related to the Hessian matrix of second-order derivatives, adaptive preconditioning is based on gradient-gradient correlations.

The parameters of a deep network are structured as a set of tensors of order two (i.e., a matrix), three, or four. For simplicity of presentation we focus on the matrix case—however our design, analysis, and implementation hold for tensors of arbitrary order. We denote the space of parameters by the matrix $W \in \mathbb{R}^{m \times n}$ and an estimate of its gradient by $G$. Full matrix Adagrad flattens $W, G$ to vectors of dimension $mn$, it thus requires $m^2 n^2$ space to store the preconditioner and $m^3 n^3$ time to perform the update. $m$ and $n$ are in the 1000's in state-of-the-art models, thus rendering full-matrix preconditioning impractical. For this reason, both AdaGrad and Adam constrain the preconditioning matrices to be diagonal. Shampoo bridges the gap between full matrix preconditioning and the diagonal version by approximating the matrices.

*The Shampoo algorithm.* We describe Shampoo in the context of the Online Convex Optimization (OCO) framework, which generalizes stochastic optimization (see, e.g., Shalev-Shwartz, 2012; Hazan, 2016). In OCO, learning progresses in rounds where on round $t$ the learner receives an input $X_t$ and then uses the parameters $W_t$ to form a prediction denoted $\hat{y}_t$. After making the prediction, the true outcome $y_t$ is revealed. The discrepancy between the true and predicted outcomes is assessed by a loss function $\ell$ which takes values in $\mathbb{R}_+$. The learner then uses the discrepancy to update the matrix to $W_{t+1}$ and prepare for the next round. For instance, the input on round $t$ can be an example $x_t \in \mathbb{R}^n$ for which the learner predicts $\hat{y} = f(W_t, x_t)$ where $f : \mathbb{R}^m \to \mathbb{R}$ and the loss is a function $\ell : \mathbb{R} \times \mathbb{R} \to \mathbb{R}_+$ such as $\ell(\hat{y}, y) = (y - \hat{y})^2$ or $\ell(\hat{y}, y) = \log(1 + \exp(-y\hat{y}))$.

Stochastic gradient methods use the gradient $G_t = \nabla_W \ell(f(W, x_t), y_t)$, thus $G_t \in \mathbb{R}^{m \times n}$ if the parameters are shaped as a matrix $W \in \mathbb{R}^{m \times n}$. For matrix-shaped parameters, Shampoo tracks two statistics over the course of its run, $L_t$ and $R_t$, which are defined as follows:

$$L_t = \epsilon I_m + \sum_{s=1}^{t} G_s G_s^\mathsf{T} ; \qquad R_t = \epsilon I_n + \sum_{s=1}^{t} G_s^\mathsf{T} G_s .$$

Note that $L_t \in \mathbb{R}^{m \times m}$ and $R_t \in \mathbb{R}^{n \times n}$. These are used to precondition the gradient and update $W$:

$$W_{t+1} = W_t - \eta \, L_t^{-1/4} G_t R_t^{-1/4} \,.$$

The primary complexity of Shampoo arises from the computation of $L_t^{-1/4}$ and $R_t^{-1/4}$, which was naively implemented using spectral decompositions (i.e., SVD).

## 3 FULL-MATRIX PRECONDITIONING: CHALLENGES

We discuss the main challenges and design choices in the development of the distributed implementation of Shampoo. These largely arose from the fact that modern accelerators are highly optimized for training using first-order optimizers, which have low computational and memory requirements. The Shampoo algorithm is computationally expensive. The extra computation in Shampoo compared to standard first-order methods is in the following steps:

- Preconditioner statistics computation: $L_t = L_{t-1} + G_t G_t^\mathsf{T}$ and $R_t = R_{t-1} + G_t^\mathsf{T} G_t$ ;
- Inverse $p$'th root computation: $L_t^{-1/4}$ and $R_t^{-1/4}$ ;
- Preconditioned gradient computation: $L_t^{-1/4} G_t R_t^{-1/4}$ .

Preconditioner statistics and gradient computations are expensive for large fully connected as well as embedding layers, we address these below. For other layers we show in Section 5 that they do not add significantly to the runtime of each step. Computing the inverse $p$'th roots is very slow—as much as 100 times the step time in some cases—and performing these without slowing down training was a key challenge in our system.

### 3.1 ALGORITHMIC CHALLENGES

Modern ML architectures often use very large embedding layers, where the longer dimension can be in the millions. For example, DLRM (Naumov et al., 2019) on Criteo-1Tb uses a vocabulary with ~186 million hash buckets, while in Transformer models (Shazeer et al., 2018) the largest layer can have up to 65536 units *per* dimension. This makes preconditioning impossible due to $O(d^2)$ memory and $O(d^3)$ computational complexity. We show how to extend Shampoo to overcome these problems; we provide proofs and convergence results in Appendix B.

*Large layers.* For embedding layers specifically, we extend the Shampoo algorithm to allow us use only one of the preconditioners, in case both preconditioners are too expensive to compute. Our choice is empirically supported by the experiments shown in Figs. 2b, 3a and 5a which suggest that there is a benefit from preconditioning one dimension of the large softmax and embedding layers with minimal increase in time. The following result allows us to choose a subset of preconditioners:

LEMMA 1. Let $G_1, \ldots, G_t \in \mathbb{R}^{m \times n}$ be matrices of rank at most $r$. Let $g_s = \mathrm{vec}(G_s)$ and define $\widehat{H}_t = \epsilon I_{mn} + \sum_{s=1}^{t} g_s g_s^\mathsf{T}$. Let $L_t, R_t$ be defined as above: $L_t = \epsilon I_m + \sum_{s=1}^{t} G_s G_s^\mathsf{T}$, $R_t = \epsilon I_n + \sum_{s=1}^{t} G_s^\mathsf{T} G_s$. Then for any $p, q > 0$ such that $1/p + 1/q = 1$, we have $\widehat{H}_t \preceq r L_t^{1/p} \otimes R_t^{1/q}$.

A consequence is that for any $p, q > 0$ such that $1/p + 1/q = 1$, the full AdaGrad preconditioned gradient $\widehat{H}_t^{-1/2} g_t$ is approximated by $(L_t^{1/p} \otimes R_t^{1/q})^{-1/2} g_t$, giving us $\widetilde{G}_t = L_t^{-1/2p} G_t R_t^{-1/2q}$. Now, by choosing $(p, q) = (1, \infty)$ and $(p, q) = (\infty, 1)$ we obtain the simple preconditioned gradients: $G_t R_t^{-1/2}$ and $L_t^{-1/2} G_t$. Theorem 3 shows that Lemma 1 can be used to prove a regret bound for this extended Shampoo in the online convex optimization setting – this provides intuitive justification for the usefulness of this approximation. We further optimize the computation of these preconditioned gradients for embedding layers by taking advantage of the sparse inputs, see details in Appendix D.

*Preconditioning blocks from large tensors.* In addition to embedding layers, large models occasionally have large fully connected layers. To reduce the computational cost of computing statistics and preconditioned gradient: we divide the tensor into blocks and treating individual block as a separate tensor. Concretely this would entail dividing tensor $W \in \mathbb{R}^{km \times kn}$, into $W_{1,1} \ldots W_{m,n}$ such that $W_{i,j} \in \mathbb{R}^{k \times k} \; \forall i, j$. Shampoo still converges in this case in the convex setting (Theorem 4), showing that the extension is justified.

LEMMA 2. Assume that $g_1, \ldots, g_t \in \mathbb{R}^{mk}$ are vectors, and let $g_i = [g_{i,1}, \ldots, g_{i,k}]$ where $g_{i,j} \in \mathbb{R}^m$. Define $\widehat{H}_t = \epsilon I_{mn} + \sum_{s=1}^{t} g_s g_s^\mathsf{T}$, and let $B_t \in \mathbb{R}^{mk \times mk}$ be the block diagonal matrix with $k$ $m \times m$ blocks, where the $j$-th block is $B_t^{(j)} = \epsilon I_m + \sum_{s=1}^{t} g_{s,j} g_{s,j}^\mathsf{T}$. Then $\widehat{H}_t \preceq k B_t$.

We performed experiments to study the effect of partitioning intermediate layers into blocks, in which we observed that the latter had minimal impact on quality of the solution while providing faster step time as well as reduced memory overheads; see Fig. 3b.

*Delayed preconditioners.* As remarked above, computing the preconditioners is the most expensive computation in every Shampoo step. In Fig. 3c we show that we can compute the preconditioners once every few hundred steps without a significant effect on the accuracy which indicates that the loss function landscape does not change significantly with each step. We observe that there is a performance/quality tradeoff here — in our experiments we set the frequency of computing preconditioners to the smallest value that does not degrade performance, i.e. the number of training steps that can be completed in the amount of time needed to compute the largest preconditioner. The only way to increase the frequency of computing preconditioners is with better hardware/software support.

## 3.2 NUMERICAL CHALLENGES

Inverse $p$'th roots (where typically $p = 2, 4, 8$) can be computed using SVD, but there are efficient iterative algorithms such as the coupled Newton iteration algorithm (Guo & Higham, 2006; Iannazzo, 2006) that can compute the inverse $p$'th root via a sequence of matrix-vector and matrix-matrix products, which are highly optimized on modern accelerators. However, our experiments suggest that on real workloads the condition numbers of the $L_t, R_t$ matrices are very large (see Fig. 6 in Appendix E) so both SVD and the coupled iteration must be run in double-precision, but this is very expensive on accelerators. We applied several further optimizations to speedup the coupled Newton iteration in our implementation; these are described in Appendix E.

## 3.3 INFRASTRUCTURAL CHALLENGES

*Heterogeneous training hardware.* Neural network accelerators are custom designed to run machine learning workloads faster and at lower cost. Accelerator design is trending towards preferring lower-precision (8-bit/16-bit) arithmetic that satisfy both of these goals on existing benchmarks. Our method demands double-precision arithmetic as described above, which makes running computation on accelerators a non-starter, and therefore we had to design the system to leverage the existing underutilized CPUs attached to the accelerators (Section 4).

*API inflexibility.* Deep learning libraries such as TensorFlow (Abadi et al., 2016) offer APIs for optimizer implementation that are well suited for first-order optimizers and for mini-batch training. Our design requires that we interact with the training loop in non-standard ways, which requires framework level changes. Our Transformer experiments were carried out in the Lingvo (Shen et al., 2019) TensorFlow framework, while BERT-Large, DRLM, as well as ResNet-50 used the MLPerf v0.7 Tensorflow baselines (Mattson et al., 2019). Experimentation required changes to the training loop such as gathering statistics at regular intervals, distributing computation across all the CPUs available in the cluster without blocking the TPU training, as well as updating the preconditioners. We anticipate that this proof-of-concept for full-matrix preconditioning will encourage the development of more flexible API's to fully utilize heterogeneous hardware.

## 4 DISTRIBUTED SYSTEM DESIGN

We present our distributed system design of the modified Shampoo algorithm. Our method is designed to run effectively on modern neural network accelerators such as TPUs (Jouppi et al., 2017) or GPUs. We first describe the standard paradigm of data parallelism used in training models on these accelerators (Dean et al., 2012). Parameters are replicated on each core of the accelerator, and each core computes forward propagation and back propagation on a sub-batch (a subset of a mini-batch, which itself is a small randomly selected subset of the training set) of input examples. These gradients are averaged across all cores via all-reduction to get the average gradient for the mini-batch. Each core uses the average mini-batch gradient to update its copy of the parameters.

All-reduction adds a barrier as all the cores need to synchronize to compute the mini-batch gradient. In Fig. 2b we measure the overhead of each of the steps on a Transformer model (Vaswani et al., 2017) described in the experiment section. We observe that the overheads from all-reduction and weight updates are a minor part ($< 5\%$) of the overall step time.

The overall design of our implementation is illustrated by the timeline in Fig. 1. As discussed in the previous section the preconditioner computation (inverse $p$th root) is expensive and requires double

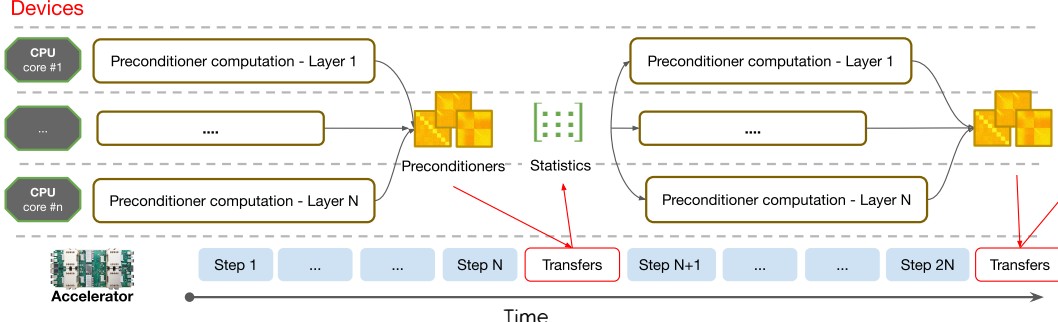

Figure 1: Timeline illustrating the design of the optimization algorithm. Preconditioner statistics ($L_t$ and $R_t$) are computed at each step by the accelerators. Preconditioners ($L_t^{1/4}$ and $R_t^{1/4}$) are only computed every $N$ steps and this computation is distributed to all available CPU cores.

precision, also we need to do this computation once every few hundred steps. These observations naturally suggested using the often underutilized CPUs on the machines to which the accelerators such as GPUs or Cloud TPUs are attached. CPUs offer double precision arithmetic but are slower than GPUs or Cloud TPUs, which makes them a perfect choice to run the preconditioner computation without adding any extra cost to the training run, as the computation is pipelined and runs asynchronously without blocking the training loop.

Preconditioners need to be computed for every layer of the network so we distribute the computation across all the CPUs that are part of the training system. As a result, the most expensive step in Shampoo adds almost nothing to the overall training time. Moreover, the computational overhead of preconditioned gradient is independent of the batch size. Thus, increasing the batch size allows us to linearly decrease the overhead making Shampoo practical for very large scale training setups. On smaller problems such as CIFAR-10, we find that our design still results in training time improvements (Appendix G.3) as preconditioner computations take very little time.

## 5 EXPERIMENTS

We compare our method against various widespread optimization algorithms for training large state-of-the-art deep models for machine translation, language modeling, recommendation systems as well as image classification. Details of the experiments are given in Appendix G and we will opensource our code before publication.

### 5.1 MACHINE TRANSLATION WITH A TRANSFORMER

We demonstrate the effectiveness of our implementation on the standard machine translation dataset from WMT'14 English to French (en→fr) with 36.3M sentence pairs. We used the state-of-the-art Transformer architecture (Vaswani et al., 2017). This architecture contains 93.3M parameters and consists of 6 layers for its encoder and decoder. Each layer is composed of 512 model dimensions, 2048 hidden dimensions, and 8 attention heads. The model makes use of a sub-word vocabulary that contains 32K word pieces (Schuster & Nakajima, 2012). The experiment was run on 32 cores of a Cloud TPU v3 Pod, and the implementation of the optimizer was carried out in the Lingvo (Shen et al., 2019) sequence to sequence modeling based on TensorFlow. Our results are shown in Fig. 2a: our algorithm achieves the same accuracy as AdaGrad or Adam in about half as many steps.

*Preconditioning of embedding and softmax layers.* Following the first extension in Section 3.1 the algorithm preconditions the large layers with only one of the preconditioners ($G_t R_t^{-1/2}$ or $L_t^{-1/2} G_t$) to make it tractable. Fig. 2b shows the increase in step time is only 6% while Fig. 3a shows that we can reduce the number of steps to convergence by ≈20%.

*Reducing overhead in fully-connected layers.* Following the second extension in Section 3.1 we ran two experiments where we partitioned fully connected layer of size [512, 2048] into two blocks of size [512, 1024] and four blocks of size [512, 512]. Our experiments show no drop in quality under this approximation with a small reduction in runtime (<3%).

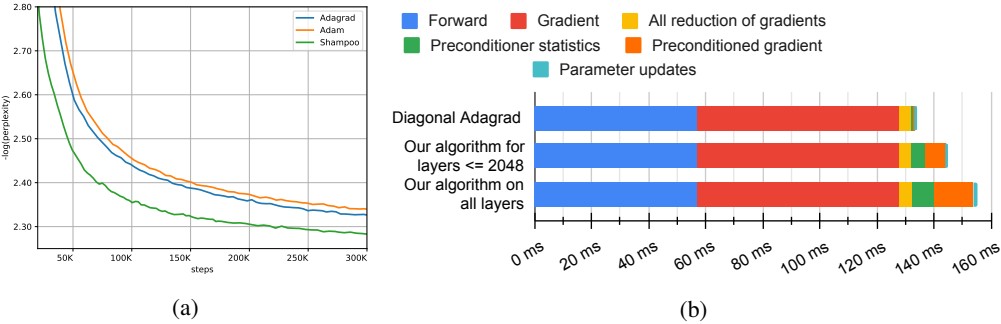

(a)          (b)

Figure 2: Results for a Transformer model on WMT'14 en→fr, trained with batch size of 1536. (a) Test log-perplexity vs. number of steps; the algorithm converges 1.95x faster in steps, while being only ≈ 16% slower per step. This allows the method to attain a particular log-perplexity in *40% less wall-time*. (b) Detailed breakdown of latency of a single step (Appendix G.6). Diagonal AdaGrad optimizer: 134ms, Shampoo: 145ms (all layers except embedding and softmax layers) and 155ms (all layers). Preconditioner computation is pipelined and distributed over CPUs, thus not adding any overhead, and transfer latency (≈100ms) is amortized over hundreds of steps.

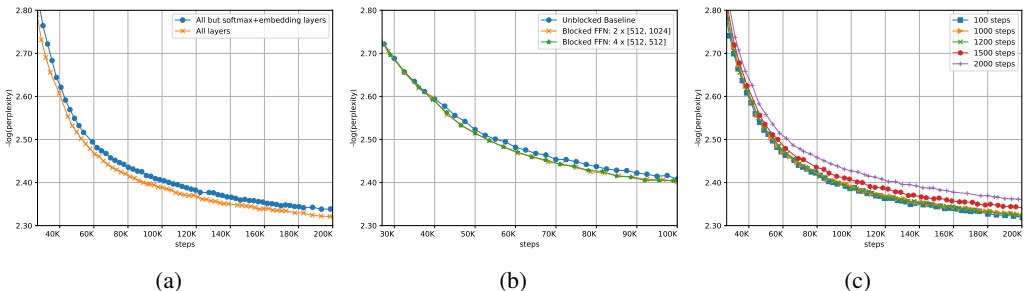

(a)          (b)          (c)

Figure 3: Impact of Shampoo extensions on WMT'14 en→fr training: (a) preconditioning applied to all layers except embedding and softmax layers, vs. applied to all layers; (b) preconditioning with fully-connected layers partitioned into sub-blocks; (c) varying interval between preconditioner updates.

## 5.2 TRANSFORMER-BIG MODEL

We also ran experiments with a larger Transformer model with 375.4M parameters, consisting of 6 layers for its encoder and decoder. Each layer is composed of 1024 model dimensions, 8192 hidden dimensions, and 16 attention heads. Results are presented in Fig. 4a where again we see an improvement in the end-to-end wall-clock time. For the softmax, embedding and the projection fully-connected layer (with 8192 hidden dimensions) we only make use of the left preconditioner. We note that step time is dominated by the preconditioned gradient computation which can be reduced by sub-blocking the layers.

*On the overhead of the optimizer.* We show the computational and memory complexity of the Shampoo extensions described in Section 3.1 in Table 2 in the appendix. We note that the overhead from computing the statistics, as well as from computing the preconditioned update for single step of training, can be further reduced by increasing the batch sizes (indeed, these overheads are independent of the batch size) as shown in Fig. 4b where the overhead dramatically reduces from 40% to 19%.

## 5.3 ADS CLICK-THROUGH RATE (CTR) PREDICTION

We trained the Deep Learning Recommendations Model (DLRM) of Naumov et al. (2019) on the terabyte Criteo click logs dataset for online advertisement click-through-rate prediction task (Criteo Labs, 2015). We compared Shampoo against the highly tuned SOTA baseline from MLPerf v0.7 training benchmarks (Wu et al., 2020). We trained the model with a batch size of 65536 for 64000 steps (1 epoch). We trained a version of the model where Shampoo is applied only to the hidden layers as well as one where we apply it for all layers. We only tune the learning rate, and keep the exact same setup as the baseline. We found that Shampoo achieves the target accuracy of 80.25% in only 30.97K steps compared to 64K steps for the baseline. Moreover, Shampoo achieves new

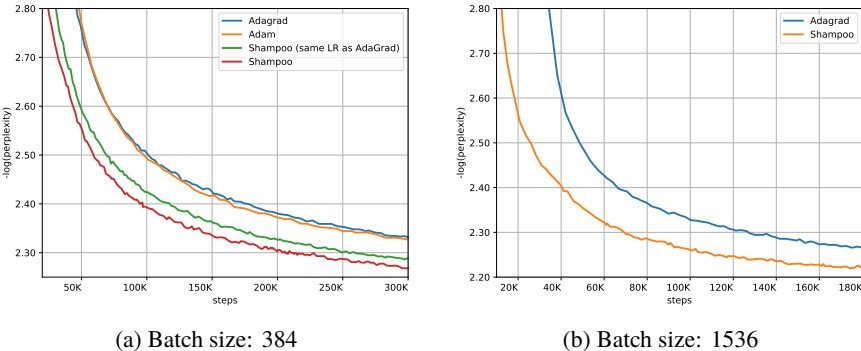

(a) Batch size: 384          (b) Batch size: 1536

Figure 4: Test log-perplexity of a Transformer-Big model on WMT'14 en→fr. (a) Shampoo converges faster than AdaGrad ($\approx 2$x faster in steps), and allows larger learning rates; due to the large overhead in step time, this results in only 30% improvement in wall-time. (b) Larger batch sizes reduce the optimizer overhead from 40% to 19%, resulting in an *end-to-end improvement of 41%* in wall-time for convergence.

state-of-the-art performance of 80.56% AUC (an $\approx 0.3\%$ improvement) on this dataset, note that an improvement of 0.1% is considered significant in this task; see Rong et al., 2020; Wang et al., 2017. Here preconditioning embedding layers further reduced the number of steps needed to reach the target accuracy from 39.96K to 30.97K.

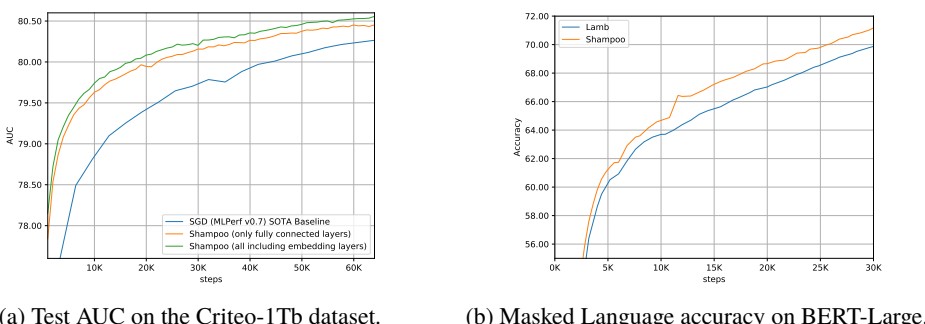

(a) Test AUC on the Criteo-1Tb dataset.   (b) Masked Language accuracy on BERT-Large.

Figure 5: (a) Shampoo reaches a target AUC of 80.25% in half as many steps with preconditioning embedding layers improving the results, and achieves a new state-of-the-art AUC of 80.56%; (b) Shampoo converges in $\approx 16\%$ fewer steps, and achieves $\approx 1\%$ higher MLM accuracy than the baseline on BERT-Large.

### 5.4 LANGUAGE MODELING

We trained BERT-Large (the Bidirectional Encoder Representation architecture of Devlin et al., 2018) for the language modeling task on the concatenation of Wikipedia and BooksCorpus, with 2.5B and 800M words respectively. BERT-Large is a large bidirectional transformer model containing 24 transformer blocks with 1024 hidden dimensions and 16 self attention heads. It has 340M parameters and is set up to jointly optimize two objectives: (a) masked language model (Masked-LM) loss where the task is to predict masked tokens based on surrounding context, and (b) next sentence prediction (NSP) loss where the task is to predict whether two given sentences are consecutive in the text. In Fig. 5b we compare our results against the current state of the art in training BERT (You et al., 2019). Models were trained with batch size 16K; in these experiments we replaced the Adam update rule in Lamb that produces the preconditioned gradient with Shampoo. Both experiments used existing well-tuned hyperparameters of the baseline.

### 5.5 IMAGE CLASSIFICATION

We trained a ResNet-50 model (He et al., 2016) on the ImageNet-2012 (Russakovsky et al., 2015) dataset and compared it against the state-of-the-art baseline using SGD+Momentum. We base our experiments off the Tensorflow baseline available from Mattson et al. (2019) where the target criteria is reaching 75.9% accuracy. See results in Table 1; in particular, we find that Shampoo reaches the target accuracy in fewer steps than the current state of the art. Tuning details are in Appendix G.4.

| OPTIMIZER | BATCH SIZE | EPOCHS | STEPS |
|---|---|---|---|
| SGD+Momentum | 4096 | 85 | 26586 |
| LARS | 4096 | 45 | 14040 |
| LARS | 32768 | 64 | 2512 |
| Shampoo | 4096 | 45 | 14040 |
| Shampoo | 16384 | 48 | 3744 |
| **Shampoo** | **32768** | **58** | **2262** |

Table 1: Epochs and steps to MLPerf target accuracy of 75.9% with a ResNet-50.

## 6 CONCLUDING REMARKS

We have presented an implementation of a second order optimizer, and demonstrated step time as well as wall time improvements on multiple large tasks in different domains — in each case our implementation performed as well or better than state-of-the-art optimizers specialized for each domain. The main point of our work is to demonstrate that second order methods implemented on a *real-world distributed setup* can be used to train state-of-the-art deep models. We hope that this work will influence future hardware accelerator design and runtime software — first order methods have received large investments in tuning, implementation, platform support and hardware tailored for them, and we believe there are several opportunities to improve the per-step time performance of second order methods as well:

- Most second order methods use symmetric matrices, but we haven't found support for typing operands as symmetric, which can reduce compute flops and storage by upto 50%.

- Several optimizations that are currently tuned towards first order methods could be extended to second order methods. For example, weight update sharding pattern matches first order methods (Xu et al., 2020) and dramatically reduces the time spent in the update step as well as memory used. This change can also be applied to Shampoo with blocked preconditioners – but we do not have support for it yet as it requires compiler level support, and is not expressible at the program layer. Currently every core must update all layers which is quite inefficient.

- Mixed precision algorithms may work for inverse pth roots and can help increase the frequency of preconditioner computation.

- Increased memory per chip can allow larger preconditioners.

- Hardware support for high-precision arithmetic in accelerators can allow more frequent preconditioner computation. The benefits of high precision arithmetic for optimization run counter to the prevailing wisdom in ML[1] which has led to the focus on low-precision formats such as bfloat16 (Wang & Kanwar, 2019).

- Hardware support for storing/packing and using upper/lower triangular matrices efficiently, as available in LAPACK.

Our hope is that these suggestions could result in innovations that would make second-order methods practical across more domains and models, especially in data limited regimes where we may not able to amortize the latency added in the data transfer between the accelerator and the CPU.

---

[1]For example, (Gupta et al., 2015) say "It is well appreciated that in the presence of statistical approximation and estimation errors, high-precision computation in the context of learning is rather unnecessary (Bottou & Bousquet, 2007)" and (Higham & Pranesh, 2019) say "... machine learning provides much of the impetus for the development of half precision arithmetic in hardware ..."

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

## A  Notation

We use lowercase letters to denote scalars and vectors, and uppercase letters to denote matrices. $\|A\|_F$ denotes the Frobenius norm of $A$, i.e., $\|A\|_F^2 = \sum_{i,j} A_{ij}^2$. $A \bullet B$ denotes the Hadamard or element-wise product of $A$ and $B$ which have the same shape, so $C = A \bullet B \iff C_{ij} = A_{ij}B_{ij}$. $D^{\odot\alpha}$ is the element-wise power, $(D^{\odot\alpha})_{ij} = D_{ij}^\alpha$.

We use $\preceq$ to denote the Loewner order: given square symmetric matrices $A, B$, we write $A \preceq B$ iff $B - A$ is positive semidefinite (PSD).

Given a symmetric PSD matrix $A$, and $\alpha \in \mathbb{R}$, $A^\alpha$ is defined as follows: let $A = UDU^\mathsf{T}$ be the singular value decomposition of $A$, where $U$ is a unitary matrix and $D$ is a diagonal matrix (with $D_{ii} \geq 0$ as $A$ is PSD), then $A^\alpha = UD^\alpha U^\mathsf{T}$, where $(D^\alpha)_{ii} = D_{ii}^\alpha$. If $\alpha < 0$, this is defined for positive definite matrices only, where $D_{ii} > 0$.

We use $\text{vec}(A)$ to denote the flattening of the $m \times n$ matrix $A$: if $A$ has rows $a_1, \ldots, a_m$, then $\text{vec}(A)$ is the $mn \times 1$ column vector $\text{vec}(A) = (a_1, \ldots, a_m)^\mathsf{T}$. $A \otimes B$ denotes the Kronecker product of two matrices $A$ and $B$, and we will use the identities $(A \otimes B)^\alpha = A^\alpha \otimes B^\alpha$ for $\alpha \in \mathbb{R}$, and $(A \otimes B)\text{vec}(C) = \text{vec}(ACB^\mathsf{T})$.

## B  Deferred Proofs

Proof (of Lemma 1). Lemma 8 in Gupta et al. (2018) shows that $\widehat{H}_t \preceq rL_t \otimes I_n$ and $\widehat{H}_t \preceq rI_m \otimes R_t$. By using Ando's inequality (Ando et al., 2004), we get

$$\widehat{H}_t \preceq r(L_t \otimes I_n)^{1/p}(I_m \otimes R_t)^{1/q}$$
$$= r(L_t^{1/p} \otimes I_n)(I_m \otimes R_t^{1/q})$$
$$= rL_t^{1/p} \otimes R_t^{1/q},$$

which concludes the proof. □

This lemma immediately allows us to prove a regret bound for Shampoo with extended exponents:

Theorem 3. Assume that the gradients $G_1, \ldots, G_T$ are matrices of rank at most $r$. Then the regret of Shampoo with extended exponents compared to any $W^\star \in \mathbb{R}^{m \times n}$ is bounded as follows,

$$\sum_{t=1}^T f_t(W_t) - \sum_{t=1}^T f_t(W^\star) \leq \sqrt{2r}D\,\text{Tr}(L_T^{\frac{1}{2p}})\,\text{Tr}(R_T^{\frac{1}{2q}}),$$

where

$$L_T = \epsilon I_m + \sum_{t=1}^T G_t G_t^\mathsf{T}, \quad R_T = \epsilon I_n + \sum_{t=0}^T G_t^\mathsf{T} G_t, \quad D = \max_{t \in [T]} \|W_t - W^\star\|_2.$$

and $1/p + 1/q = 1, p, q \geq 1$.

Proof. The proof follows the proof of Theorem 7 in Gupta et al. (2018). Let $H_t = L_t^{\frac{1}{2p}} \otimes R_t^{\frac{1}{2q}}$. Then the update rule of the extended Shampoo algorithm is equivalent to $w_{t+1} = w_t - \eta H_t^{-1} g_t$. Since $0 \preceq L_1 \preceq \ldots \preceq L_T$ and $0 \preceq R_1 \preceq \ldots \preceq R_T$, standard properties of the Kronecker product and the operator monotonicity of the function $x \mapsto x^\alpha$ for $\alpha \leq 1$ (an immediate consequence of Ando's inequality) ensure that $0 \preceq H_1 \preceq \ldots \preceq H_T$.

Following the aforementioned proof, we have the regret bound

$$\sum_{t=1}^T f_t(W_t) - \sum_{t=1}^T f_t(W^\star) \leq \frac{D^2}{2\eta}\,\text{Tr}(H_T) + \frac{\eta}{2}\sum_{t=1}^T \|g_t\|_{H_t^*}^2,$$

where $D = \max_t \|W_t - W^\star\|_2$. Define $g_t = \text{vec}(G_t)$ and $\widehat{H}_t = (\epsilon I_m + \sum_{s=1}^t g_s g_s^\mathsf{T})^{1/2}$, then Lemma 1 shows that $\widehat{H}_t \preceq \sqrt{r}H_t$, using operator monotonicity. Using this equation twice, along with Equation (6) from the proof of Theorem 7, we have

$$\sum_{t=1}^T \|g_t\|_{H_t^*}^2 \leq \sqrt{r}\sum_{t=1}^T \|g_t\|_{\widehat{H}_t^*}^2 \leq 2\sqrt{r}\,\text{Tr}(\hat{H}_T) \leq 2r\,\text{Tr}(H_T).$$

This gives us

$$\sum_{t=1}^{T} f_t(W_t) - \sum_{t=1}^{T} f_t(W^\star) \le \frac{D^2}{2\eta} \operatorname{Tr}(H_T) + \eta r \operatorname{Tr}(H_T).$$

Setting $\eta = D/\sqrt{2r}$ and observing that $\operatorname{Tr}(H_t) = \operatorname{Tr}(L_t^{1/2p}) \operatorname{Tr}(R_t^{1/2q})$ gives us the required bound. $\quad\square$

PROOF (OF LEMMA 2). Let $x \in \mathbb{R}^{mk}$, and $x = [x_1, x_2, \ldots, x_k]$, where $x_j \in \mathbb{R}^m$. Then

$$x^\mathsf{T} \widehat{H}_t x = \epsilon \|x\|_2^2 + \sum_{s=1}^{t} x^\mathsf{T} g_s g_s^\mathsf{T} x = \epsilon \|x\|_2^2 + \sum_{s=1}^{t} (g_s^\mathsf{T} x)^2 = \epsilon \|x\|_2^2 + \sum_{s=1}^{t} \left( \sum_{j=1}^{k} g_{s,j}^\mathsf{T} x_j \right)^2$$

$$\le k\epsilon \|x\|_2^2 + k \sum_{s=1}^{t} \sum_{j=1}^{k} (g_{s,j}^\mathsf{T} x_j)^2 = k \sum_{j=1}^{k} \left( \epsilon \|x_j\|_2^2 + \sum_{s=1}^{t} x_j^\mathsf{T} g_{s,j} g_{s,j}^\mathsf{T} x_j \right)$$

$$= k \sum_{j=1}^{k} x_j^\mathsf{T} \left( \epsilon I_m + \sum_{s=1}^{t} g_{s,j} g_{s,j}^\mathsf{T} \right) x_j = k \sum_{j=1}^{k} x_j^\mathsf{T} B_t^{(j)} x_j = k x^\mathsf{T} B_t x.$$

Here we used the inequality $\left( \sum_{j=1}^{k} \alpha_j \right)^2 \le k \sum_{j=1}^{k} \alpha_j^2$, which follows from the convexity of $x \mapsto x^2$ (or from the fact that variance of a random variable is non-negative). $\quad\square$

This lemma once again allows us to prove a regret bound, exactly following the proof of the regret bound above:

THEOREM 4. Assume that the gradients are $g_1, \ldots, g_T \in \mathbb{R}^{mk}$, and let $g_i = [g_{i,1}, \ldots, g_{i,k}]$ where $g_{i,j} \in \mathbb{R}^m$. Then the regret of Shampoo with blocking compared to any $w^\star \in \mathbb{R}^{mk}$ is bounded as follows:

$$\sum_{t=1}^{T} f_t(w_t) - \sum_{t=1}^{T} f_t(w^\star) \le \sqrt{2k} D \sum_{j=1}^{k} \operatorname{Tr}\left( \left( \epsilon I_m + \sum_{t=1}^{T} g_{t,j} g_{t,j}^\mathsf{T} \right)^{\frac{1}{2}} \right).$$

The two regret bounds can be combined to show that Shampoo with both extensions also converges.

## C  COMPARISON WITH K-FAC

K-FAC is a natural gradient algorithm, and approximates the curvature of the loss using the Fisher Information Matrix:

$$\mathbf{F} = \mathop{\mathbb{E}}_{p(x|\theta)} \left[ \nabla \log p(x|\theta) \nabla \log p(x|\theta)^\mathsf{T} \right] = \mathop{\mathbb{E}}_{p(x|\theta)} \left[ g_{p(x|\theta)} g_{p(x|\theta)}^\mathsf{T} \right].$$

For a fully connected layer with $W \in \mathbb{R}^{m \times n}$, where $Wx = s$, the gradient for the layer $G_t \in \mathbb{R}^{m \times n}$ can be written via the chain rule as $G_t = \nabla_s \ell(s_t, y_t) x^\mathsf{T}$ and in vectorized form as: $\nabla_s \ell(s_t, y_t) \otimes x$. We can then write the Fisher information matrix as:

$$\mathbf{F} = \mathop{\mathbb{E}}_{p(x|\theta)} \left[ (\nabla_s \ell(s_t, y_t) \otimes x)(\nabla_s \ell(s_t, y_t) \otimes x)^\mathsf{T} \right]$$

$$= \mathop{\mathbb{E}}_{p(x|\theta)} \left[ (\nabla_s \ell(s_t, y_t) \nabla_s \ell(s_t, y_t)^\mathsf{T}) \otimes (x_t x_t^\mathsf{T}) \right].$$

Assuming independence between $\nabla_s \ell(s_t, y_t)$ and $x$, K-FAC rewrites the Fisher in tractable form as:

$$\mathbf{F} \approx \mathbb{E} \left[ (\nabla_s \ell(s_t, y_t) \nabla_s \ell(s_t, y_t)^\mathsf{T}) \right] \otimes \mathbb{E} \left[ x_t x_t^\mathsf{T} \right].$$

If we let $D = \mathbb{E} \left[ (\nabla_s \ell(s_t, y_t) \nabla_s \ell(s_t, y_t)^\mathsf{T}) \right]$ and $X = \mathbb{E} \left[ x_t x_t^\mathsf{T} \right]$, the update rule then becomes:

$$W_{t+1} \approx W_t - \eta D^{-1} G_t X^{-1}.$$

We note some of the differences and similarities between the two updates here. KFAC preconditioners use exponent of $-1$ (as original Fisher is inverted) whereas Shampoo uses $-1/2p$ where $p$ is the rank of the tensor. KFAC computes statistics based on gradients with labels sampled from the model's

predictive distribution (hence requiring strictly more computation) where as Shampoo relies on the gradient of the mini-batch.

Now we can compute each term in the Shampoo preconditioners as:

$$G_t G_t^\mathsf{T} = \nabla_s \ell(s_t, y_t) x_t^\mathsf{T} x_t \nabla_s \ell(s_t, y_t)^\mathsf{T} = \|x_t\|_2^2 \nabla_s \ell(s_t, y_t) \nabla_s \ell(s_t, y_t)^\mathsf{T};$$
$$G_t^\mathsf{T} G_t = x_t \nabla_s \ell(s_t, y_t)^\mathsf{T} \nabla_s \ell(s_t, y_t) x_t^\mathsf{T} = \|\nabla_s \ell(s_t, y_t)\|_2^2 x_t x_t^\mathsf{T}.$$

Dividing by the scale, and taking expectations on both sides:

$$\mathbb{E}\left[\frac{G_t G_t^\mathsf{T}}{\|x_t\|_2^2}\right] = \mathbb{E}\left[\nabla_s \ell(s_t, y_t) \nabla_s \ell(s_t, y_t)^\mathsf{T}\right] = D;$$

$$\mathbb{E}\left[\frac{G_t^\mathsf{T} G_t}{\|\nabla_s \ell(s_t, y_t)\|_2^2}\right] = \mathbb{E}\left[x_t x_t^\mathsf{T}\right] = X.$$

This shows that K-FAC preconditioners are closely related to Shampoo preconditioners, especially when one uses the empirical Fisher (Kunstner et al., 2019).

The main difficulty in implementing K-FAC on a model is that current optimizer APIs make it difficult to send additional information such as $\|x_t\|_2^2, \|\nabla_s \ell(s_t, y_t)\|_2^2$ to the optimizer, so K-FAC implementations have to register the structure of each layer. Moreover, due to the dependence of K-FAC on the structure of the network, it is difficult to implement standard operators like batch norm, weight norm, layer norm, etc., which are prevalent in the tasks and models we considered. For example, if we write a fully connected layer with weight norm as $s = Wx/\|W\|$, then the gradient

$$G_t = \frac{1}{\|W\|} \nabla_s \ell(s_t, y_t) x^\mathsf{T} - \frac{\nabla_s \ell(s_t, y_t)^\mathsf{T} W x}{\|W\|^3} W,$$

so rewriting $\mathbb{E}[\mathrm{vec}(G_t) \mathrm{vec}(G_t)^\mathsf{T}]$ as a Kronecker product is not an easy task.

The similarity between K-FAC and Shampoo preconditioners also allows us to use techniques explored by the K-FAC community for Shampoo. One of the extensions for KFAC is the E-KFAC algorithm (George et al., 2018) which constructs a better approximation of the Fisher matrix by using the eigenbasis computed from the Kronecker approximation, but rescaling the eigenvalues to match the diagonal of the Fisher matrix in this eigenbasis. This method produces a provably better approximation, and can immediately be applied to Shampoo too with a simple modification:

Let $\hat{H}_t \approx L_t^{1/2} \otimes R_t^{1/2}$. Let the singular value decompositions of the factors be $L_t^{1/2} = UDU^\mathsf{T}$ and $R_t^{1/2} = VD'V^\mathsf{T}$. The $L_t^{1/2} \otimes R_t^{1/2} = (U \otimes V)(D \otimes D')(U \otimes V)^\mathsf{T}$. Now the EKFAC correction replaces $D \otimes D'$ by the optimal diagonal

$$\Lambda = \mathrm{diag}((U \otimes V)^\mathsf{T} \hat{H}_t (U \otimes V))$$

$$= \epsilon I + \sum_{s=1}^{t} \mathrm{diag}((U \otimes V)^\mathsf{T} \mathrm{vec}(G_s) \mathrm{vec}(G_s)^\mathsf{T} (U \otimes V))$$

$$= \epsilon I + \sum_{s=1}^{t} \mathrm{diag}(\mathrm{vec}(U^\mathsf{T} G_s V) \mathrm{vec}(U^\mathsf{T} G_s V)^\mathsf{T})$$

$$= \epsilon I + \sum_{s=1}^{t} \mathrm{vec}(U^\mathsf{T} G_s V)^{\odot 2},$$

Thus we can approximately compute $\Lambda_{t+1} \approx \Lambda_t + (U^\mathsf{T} G_t V)^{\odot 2}$, and the new update becomes: $W_{t+1} = W_t - \eta_t U(\Lambda_t^{-1/2} \bullet (U^\mathsf{T} G_t V))V^\mathsf{T}$. This technique does have the disadvantage that it requires computing the singular value decompositions (which we already observed are much slower than coupled Newton iterations), and doubles the number of matrix multiplications in the preconditioned gradient computation. At this time our experiments did not show significant improvements over the standard Shampoo implementation, but we plan to explore this further.

## D    SHAMPOO FOR EMBEDDING LAYERS

In modern networks, embedding layers are usually very large, and even computing the left preconditioner as described in Section 3.1 can be prohibitively expensive. However we can take advantage

of the fact that the inputs to the network are very sparse, and use this to reduce the computation significantly.

Let our input example to such a network consist of a set of categorical features: each feature such as user language, user country etc consists of one out of a set of options. Then the output of the embedding layer is the concatenation of the embeddings for each such feature. If the embeddings are of width $d$ and there are $N$ such embeddings, then the embedding layer is $W \in \mathbb{R}^{d \times N}$. The input can be represented as $x \in \mathbb{R}^{N \times m}$, where $m$ is the number of categorical features, and each column is one-hot: if the $k$-th feature is $x(k)$, then $x_{jk} = \delta_{j,x(k)}$. The output of the layer is $y = Wx$.

Now $G = \nabla_W \ell = \nabla_y \ell \, x^\mathsf{T}$, so $GG^\mathsf{T} = \nabla_y \ell \, x^\mathsf{T} x \, \nabla_y \ell^\mathsf{T}$. But $x^\mathsf{T} x = \mathbf{I}_m$, so $GG^\mathsf{T} = \nabla_y \ell \, \nabla_y \ell^\mathsf{T}$. Thus we can compute the preconditioner for $W$ by computing it on the output of the embedding layer, and this is a much smaller computation since $y$ is of dimension $b \times m$, this computation is $O(d^2 m)$ rather than $O(d^2 N)$. Note that sparse multiplication would also be $O(d^2 m)$, but accelerators usually implement sparse operations by densifying the tensors.

If each column of $x$ is multi-hot, as is the case when the features are words and their embeddings are averaged, $x^\mathsf{T} x$ is a diagonal matrix, where each diagonal entry is a function of the number of ones in each column of $x$. Computing $GG^\mathsf{T} = \nabla_y \ell (x^\mathsf{T} x) \nabla_y \ell^\mathsf{T}$ is still $O(d^2 m) \ll O(d^2 N)$.

# E   A COUPLED NEWTON ITERATION FOR COMPUTATION OF INVERSE $p$-TH ROOTS

The Newton method for solving the matrix equation $X^{-p} - A = 0$ produces the iteration $X_{k+1} = \frac{1}{p}[(p+1)X_k - X_k^{p+1} A]$, where we take $X_0 = \frac{1}{c} I$. This iteration satisfies $X_k \to A^{-1/p}$ as $k \to \infty$, but it is not numerically stable. Introducing the matrix $M_k = X_k^p A$, we get

$$X_{k+1} = X_k \left( \frac{(p+1)I - M_k}{p} \right), \qquad X_0 = \frac{1}{c} I,$$

and

$$M_{k+1} = X_{k+1}^p A = \left( \frac{(p+1)I - M_k}{p} \right)^p X_k^p A = \left( \frac{(p+1)I - M_k}{p} \right)^p M_k, \qquad M_0 = \frac{1}{c^p} A,$$

since $X_k, M_k$ and $A$ commute with each other. This is the coupled Newton iteration for computing inverse $p$-th roots, and was shown to be numerically stable in (Guo & Higham, 2006; Iannazzo, 2006).

We implemented the following optimizations to the coupled Newton iteration method:

- *Warm Start*: The coupled Newton iteration to compute $G^{-1/p}$ starts with $X = I, M = G$ and maintains the invariant $M = X^p G$ while driving $M \to I$, resulting in $X \to G^{-1/p}$. We need to find the $p$-th root of a sequence $G_t$, so we instead set $X = G_t^{-1/p}, M = X^p G_{t+1}$; since the difference between $G_t$ and $G_{t+1}$ is small, this ensures that $M$ is already close to $I$. In our experiments warmstart improves convergence (by upto 4x fewer steps).

- *Projecting top singular values*: In practice our $G_t$ matrices have large condition numbers, which sometimes leads to inaccurate results. As a rule of thumb, computing $G^{-1/p}$ leads to a loss of $\log_2(\frac{1}{p}\kappa(G))$ bits of precision (Overton, 2001), where $\kappa(G)$ is the condition number of the $G$. However we also find that usually there is a sharp falloff within the first few singular values, so in order to reduce the condition number, we project away the largest singular values, since these are the least important after taking inverses. We find the top-$k$ singular values $\lambda_1, \ldots, \lambda_k$ and their associated singular vectors using a standard iterative method, and replace each with $\lambda_{k+1}$. This produces a better approximation than adding $\epsilon I$ to each $G_t$: the latter can wash out the smallest (and most crucial) singular values, see Fig. 7 where the smallest singular value for a layer can be as small as $10^{-10}$ to $10^{-6}$ during the course of optimization.

- *Dynamic tuning of projection*: We dynamically tune the number of singular values we need to project in the previous step, by computing the condition number $\kappa(G_t)$ and using it to estimate the smallest singular value of $G_{t+1}$ as $\lambda_{\max}(G_{t+1})/\kappa(G_t)$. We then keep projecting out singular values of $G_{t+1}$ until we get an acceptable condition number.

- *Correcting for projection*: If $G_t = \sum_i \lambda_i \mathbf{v}_i \mathbf{v}_i^\mathsf{T}$, then $G_t^{-1/p} = \sum_i \lambda_i^{-1/p} \mathbf{v}_i \mathbf{v}_i^\mathsf{T}$. Projection above means replacing $\lambda_1, \ldots \lambda_k$ by $\lambda_{k+1}$, but since we have already computed the corresponding $\mathbf{v}_1, \ldots \mathbf{v}_k$, we correct the approximate $p$-th root by adding $\sum_{i=1}^k (\lambda_i^{-1/p} - \lambda_{k+1}^{-1/p}) \mathbf{v}_i \mathbf{v}_i^\mathsf{T}$. This is a small effect, but adding it is a straightforward modification (details deferred to Appendix E).

---

**Algorithm I** A coupled Newton iteration procedure for computing inverse $p$-th roots of a PSD matrix, with warm start and singular value projection

```
 1: procedure MAXSV(G)
 2:     Parameters: ϵ > 0, n_step
 3:     v ∈ ℝⁿ, where G ∈ ℝⁿˣⁿ
 4:     i = 0, error = ∞, λ = 0
 5:     while i < n_step and error > ϵ do
 6:         v̂ = v/‖v‖
 7:         v = Gv̂
 8:         λ_old = λ; λ = v̂ᵀv
 9:         error = |λ − λ_old|; i = i + 1
10:     return λ, v/‖v‖
11:
12: procedure PROJECT(G, κ (optional), κ_d (optional), n_proj (optional))
13:     i = 0
14:     Δ = 0
15:     λ, v = MAXSV(G)
16:     λ_max = λ
17:     while λ > (κ_d/κ)λ_max or i < n_proj do
18:         G = G − λvvᵀ
19:         Δ = Δ + vvᵀ
20:         λ, v = MaxSV(G)
21:         i = i + 1
22:     return G + λΔ
23:
24: procedure COUPLEDITERATION(G, p ∈ ℕ, X (optional), κ (optional))
25:     Parameters: ϵ > 0, κ_d, n_proj
26:     Outputs: G^{−1/p}
27:     G = PROJECT(G, κ, κ_d, n_proj)
28:     α = −1/p
29:     if X is provided then
30:         M = XᵖG
31:     else
32:         z = (1+p)/(2‖G‖_F)
33:         X = (1/zᵅ)I
34:         M = zG
35:     while ‖M − I‖_∞ > ϵ do
36:         M₁ = (1 − α)I + αM
37:         X = XM₁
38:         M = M₁ᵖM
39:     return X
```

$$
\begin{aligned}
&1: \textbf{procedure } \textsc{MaxSV}(\mathbf{G}) \\
&2: \quad \textbf{Parameters: } \epsilon > 0, n_{\text{step}} \\
&3: \quad \mathbf{v} \in \mathbb{R}^n, \text{ where } \mathbf{G} \in \mathbb{R}^{n \times n} \\
&4: \quad i = 0, \text{error} = \infty, \lambda = 0 \\
&5: \quad \textbf{while } i < n_{\text{step}} \text{ and error} > \epsilon \textbf{ do} \\
&6: \qquad \hat{\mathbf{v}} = \mathbf{v}/\|\mathbf{v}\| \\
&7: \qquad \mathbf{v} = \mathbf{G}\hat{\mathbf{v}} \\
&8: \qquad \lambda_{\text{old}} = \lambda; \lambda = \hat{\mathbf{v}}^\mathsf{T}\mathbf{v} \\
&9: \qquad \text{error} = |\lambda - \lambda_{\text{old}}|; i = i + 1 \\
&10: \quad \textbf{return } \lambda, \mathbf{v}/\|\mathbf{v}\| \\
&12: \textbf{procedure } \textsc{Project}(\mathbf{G}, \kappa \text{ (optional)}, \kappa_d \text{ (optional)}, n_{\text{proj}} \text{ (optional)}) \\
&13: \quad i = 0 \\
&14: \quad \Delta = \mathbf{0} \\
&15: \quad \lambda, \mathbf{v} = \textsc{MaxSV}(\mathbf{G}) \\
&16: \quad \lambda_{\max} = \lambda \\
&17: \quad \textbf{while } \lambda > \frac{\kappa_d}{\kappa}\lambda_{\max} \text{ or } i < n_{\text{proj}} \textbf{ do} \\
&18: \qquad \mathbf{G} = \mathbf{G} - \lambda\mathbf{v}\mathbf{v}^\mathsf{T} \\
&19: \qquad \Delta = \Delta + \mathbf{v}\mathbf{v}^\mathsf{T} \\
&20: \qquad \lambda, \mathbf{v} = \text{MaxSV}(\mathbf{G}) \\
&21: \qquad i = i + 1 \\
&22: \quad \textbf{return } \mathbf{G} + \lambda\Delta \\
&24: \textbf{procedure } \textsc{CoupledIteration}(\mathbf{G}, p \in \mathbb{N}, \mathbf{X} \text{ (optional)}, \kappa \text{ (optional)}) \\
&25: \quad \textbf{Parameters: } \epsilon > 0, \kappa_d, n_{\text{proj}} \\
&26: \quad \textbf{Outputs: } \mathbf{G}^{-1/p} \\
&27: \quad \mathbf{G} = \textsc{Project}(\mathbf{G}, \kappa, \kappa_d, n_{\text{proj}}) \\
&28: \quad \alpha = -\frac{1}{p} \\
&29: \quad \textbf{if } \mathbf{X} \text{ is provided } \textbf{then} \\
&30: \qquad \mathbf{M} = \mathbf{X}^p\mathbf{G} \\
&31: \quad \textbf{else} \\
&32: \qquad z = \frac{1+p}{2\|\mathbf{G}\|_F} \\
&33: \qquad \mathbf{X} = \frac{1}{z^\alpha}\mathbf{I} \\
&34: \qquad \mathbf{M} = z\mathbf{G} \\
&35: \quad \textbf{while } \|\mathbf{M} - \mathbf{I}\|_\infty > \epsilon \textbf{ do} \\
&36: \qquad \mathbf{M}_1 = (1 - \alpha)\mathbf{I} + \alpha\mathbf{M} \\
&37: \qquad \mathbf{X} = \mathbf{X}\mathbf{M}_1 \\
&38: \qquad \mathbf{M} = \mathbf{M}_1^p\mathbf{M} \\
&39: \quad \textbf{return } \mathbf{X}
\end{aligned}
$$

## F  IMPLEMENTATION DETAILS OF SHAMPOO

Our implementation of the Shampoo algorithm for fully-connected layers is described in Algorithm II. The algorithm can use heavy-ball momentum for its updates, as well an exponential moving average over the preconditioners, like Adam. The configuration parameter $\tau_1$ denotes the number of steps between subsequent fetches of the latest available preconditioner by the accelerator. $\tau_1$ must be set sufficiently high so that there is enough time for the CPU to complete the computation of the preconditioner asynchronously and pipeline it efficiently, but otherwise its setting does not have a significant effect on convergence. The configuration parameter $\tau_2$ (default value = 1) determines the frequency of gathering gradient statistics - we update $L_t, R_t$ every $\tau_2$ steps only for efficiency.

### F.1  COMPUTATION COST OF SHAMPOO

We capture the computational and memory complexity under various schemes described in Section 3.1 of handling large layers in Table 2.

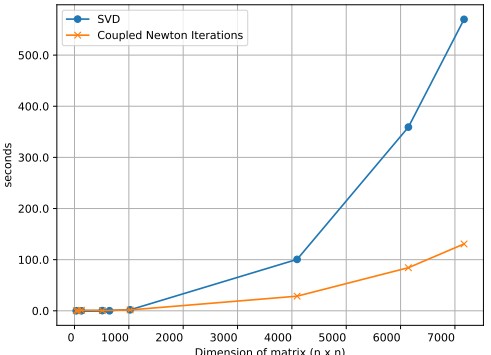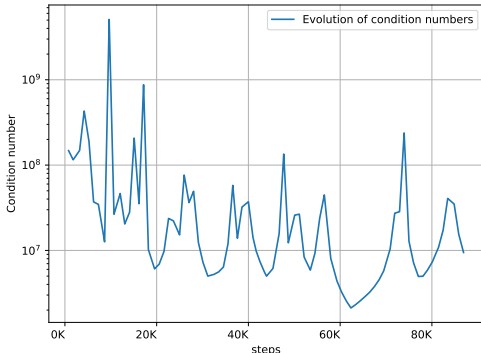

Figure 6: Benchmarks on computing inverse-pth root for statistics of varying dimensions (left), and the condition numbers for $L_t$ of a layer in the transformer model over time (right). We find that the coupled Newton iteration method can effectively utilize the CPUs and give large walltime improvements compared to SVD (that relies on bidiagonal divide-and-conquer). These were measured without warmstart which provides additional speedup of upto 4x by reducing the number of iterations to the solution. These were measured on Intel Skylake CPUs. Note that since $\sim \log_2(\frac{1}{p}\kappa(L_t))$ bits of precision are lost in computing $p$-th roots, 64-bit arithmetic becomes necessary.

---

**Algorithm II** Sketch of the Shampoo algorithm

1: **parameters:** learning rate $\eta_t$, momentum: $\beta_1$, $\beta_2$
2: **for** $t = 1, \ldots, T$ **do**
3:      Receive stochastic gradients $G_t$ for each layer
4:      **if** $t \% \tau_2 = 0$ **then**
5:          **if** $\beta_2 < 1$ **then**
6:              $L_t \leftarrow \beta_2 \, L_{t-\tau_2} + (1 - \beta_2) \, G_t G_t^{\mathsf{T}}$
7:              $R_t \leftarrow \beta_2 \, R_{t-\tau_2} + (1 - \beta_2) \, G_t^{\mathsf{T}} G_t$
8:          **else**
9:              $L_t \leftarrow L_{t-\tau_2} + G_t G_t^{\mathsf{T}}$
10:             $R_t \leftarrow R_{t-\tau_2} + G_t^{\mathsf{T}} G_t$
11:      $D_t \leftarrow D_{t-1} + G_t \bullet G_t$
12:      $M_t \leftarrow \beta_1 \, M_{t-1} + (1 - \beta_1) \, D_t^{\odot -1/2} \bullet G_t$
13:      **if** $t \% \tau_1 = 0$ **then**
14:          Gather preconditioners $L_{(t-\tau_1)}^{-1/4}, R_{(t-\tau_1)}^{-1/4}$ from CPUs
15:          Send $L_t, R_t$ to CPU host to compute $L_t^{-1/4}, R_t^{-1/4}$
16:      **if** $t > \tau_1$ **then**
17:          $P_t \leftarrow \beta_1 P_{t-1} + (1 - \beta_1) \, L_t^{-1/4} G_t R_t^{-1/4}$
18:          $\eta_t \leftarrow \eta_0 \|M_t\|_F / \|P_t\|_F$
19:          $W_t = W_{t-1} - \eta_t P_t$
20:      **else**
21:          $\eta_t \leftarrow \eta_0$
22:          $W_t = W_{t-1} - \eta_t M_t$

---

## G    FURTHER DETAILS ON EXPERIMENTS

*Layer wise learning rates.* As seen in Fig. 7 the step size scale for each layer is dependent on the operator norm of the preconditioners (inverse-pth root of the smallest singular value of the statistics

| TYPE | COMPUTATION | MEMORY |
|---|---|---|
| All preconditioner $W_t$: $[n, m]$ | $O(n^2 m + m^2 n)$ | $O(n^2 + m^2)$ |
| Left only preconditioner for $W_t$: $[n, m]$ | $O(n^2 m)$ | $O(n^2)$ |
| Preconditioner: block size $b$ | $O(mnb)$ | $O(mn)$ |

Table 2: Computational and memory complexity of variants of Shampoo.

matrix) has large spread in its range which results in optimization instabilities in practice. Moreover, as statistics as well as preconditioner computation are amortized across many steps the norm does not grow at every step. Hence, we rely on a learning rate schedule based on the update directions of a well tuned first order optimizer (in our experiments we use diagonal AdaGrad for Transformers in machine translation, as well as Criteo, layer-wise scaling heuristic proposed in LARS/LAMB optimizer, where each layer's learning rate is set to be $\left\|W_t\right\|_F / \left\|G_t\right\|_F$ for BERT and ResNet training. For example, when used with diagonal AdaGrad: Shampoo is used to determine the direction of the update, and AdaGrad to determine its magnitude.

This procedure termed Grafting in (Agarwal et al., 2020) allows us to bootstrap a reasonable learning rate schedule for a specific problem that is well tuned, and study the effect of preconditioned gradient directions in isolation. The weight matrix $W_t$ is updated as $W_t = W_{t-1} - A_t \hat{\mathbf{S}}_{\mathbf{t}}$, where:

$$D_t = \sum_{s=1}^{t} G_s \bullet G_s; \quad A_t = \eta_0 \left\|D_t^{\odot -1/2} \bullet G_t\right\|_F \qquad \text{(Adagrad magnitude)}$$

$$\hat{\mathbf{S}}_{\mathbf{t}} = \frac{L_t^{-1/4} G_t R_t^{-1/4}}{\left\|L_t^{-1/4} G_t R_t^{-1/4}\right\|_F} \qquad \text{(Shampoo direction)}.$$

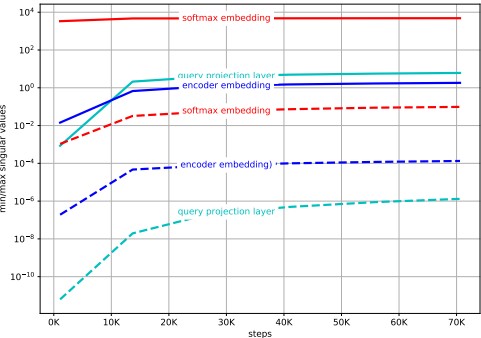

Figure 7: Minimum (dashed) and maximum (solid) singular values for statistics matrices of the embedding, softmax and intermediate attention query projection layers.

### G.1 Transformer model on WMT'14 en→fr

For all optimizers, we make use of a warmup schedule where the learning rate is increased from 0.0 to $\eta$ over 40k steps. For the smaller transformer experiments, we use a quadratic warmup, and for the larger transformer experiments we use a linear warmup. We found that quadratic warmup improves all optimizers equally and provides a better log-perplexity. For the Adam optimizer experiments, we use a learning rate decay schedule of the form $\eta_t = \eta\sqrt{d/t}$, following the suggestion of Vaswani et al. (2017). For the smaller Transformer experiments, we tuned the hyperparameters for each algorithm over 100 trials. We took the best settings for the momentum and second-moment parameters, and tuned the learning rates until either the model became unstable, or did not increase performance. For Shampoo, we used a per layer learning rate derived from AdaGrad (see Appendix G for details), and found that for the exact same hyperparameter settings as AdaGrad, Shampoo provides a modest improvement in performance. Moreover, Shampoo allows for larger learning rates than AdaGrad does, as shown in Fig. 4a.

### G.2 Step time for BERT-Large

Our current implementation showed a 14% increase in step time for BERT-Large, nearly wiping out all the gains from reduced number of steps (16%). We note that due amount of resources it would require to tune BERT, we used Shampoo with exact same hyper-parameters as LAMB with grafting to understand the effect of preconditioner. Moreover, step time can be optimized considerably as the current implementation is not heavily optimized. For example, larger batch sizes help amortize the preconditioning overhead, and reduce overall wall time to reach the same accuracy. Furthermore,

| EXPERIMENT (TPU CORES) | OPTIMIZER | BATCH | OPTIMIZER PARAMETERS | WARMUP |
|---|---|---|---|---|
| Transformer (32) | Adam | 1536 | $\eta = 0.000225, \beta_1 = 0.9, \beta_2 = 0.98$ | 40k steps |
| | Adagrad | 1536 | $\eta = 0.125, \beta_1 = 0.95$ | 40k steps |
| | Shampoo | 1536 | $\eta = 0.225, \beta_1 = 0.95, \kappa = 500$ | 40k steps |
| | | | $\tau_1 = 1000, \tau_2 = 1$ | |
| Transformer-Big (32) | Adam | 384 | $\eta = 0.000154, \beta_1 = 0.9, \beta_2 = 0.999$ | 40k steps |
| | Adagrad | 384 | $\eta = 0.03, \beta_1 = 0.9$ | 40k steps |
| | Shampoo | 384 | $\eta = 0.06, \beta_1 = 0.9, \kappa = 500$ | 40k steps |
| | | | $\tau_1 = 1000, \tau_2 = 1$ | |
| Transformer-Big (32) | Adagrad | 1536 | $\eta = 0.06, \beta_1 = 0.9$ | 40k steps |
| | Shampoo | 1536 | $\eta = 0.08, \beta_1 = 0.9, \kappa = 500$ | 40k steps |
| | | | $\tau_1 = 1000, \tau_2 = 1$ | |
| Bert-Large (256) | LAMB | 16384 | $\eta = 0.0060 \; \beta_1 = 0.9, \beta_2 = 0.999$ | 6.4k steps |
| | Shampoo | 16384 | $\eta = 0.0060 \; \beta_1 = 0.9, \beta_2 = 0.999,$ | 6.4k steps |
| | | | $\lambda_2 = 10^{-2}, \tau_1 = 400, \tau_2 = 10$ | |
| | | | Block size: 1024 | |
| DLRM (32) | SGD | 65536 | $\eta = 0.1$, poly decay(p=2) at 38k steps | 2k steps |
| | Shampoo | 65536 | $\eta = 0.1$ poly decay(p=2) at 38k steps | |
| | | | $\beta_1 = 0.9, \tau_1 = 999, \tau_2 = 10$ | 2k steps |
| | (w/ embd) | 65536 | $\eta_{embd} = 0.31$ | |

Table 3: Hyperparameter setup used in our experiments.

in our current implementation, all TPU cores compute all the preconditioning statistics and the preconditioned gradients, which involves over a hundred $1024 \times 1024$ matrix multiplications. This repeated work can be avoided by cross-replica sharding of weight update (Xu et al., 2020), which distributes this computation across cores, and should save at least half the step time overhead.

## G.3 CIFAR-10

We train a ResNet-50 model on CIFAR-10 (Krizhevsky et al., 2009) with 2 cores of CloudTPU-v2 at batch size 2048. Our baseline achieves 93.45% accuracy at 300 epochs, where as Shampoo reaches the same accuracy in 143 epochs. We see an overall training time reduction of 42% (1428 seconds to 827 seconds). As it is a smaller problem, the time taken for preconditioner inverse computation for the largest preconditioning matrix is less than 1ms on the CPU. We use a total of 8 CPU cores to run these inverses.

## G.4 IMAGENET

For SGD with Momentum, the learning rate is warmed up over the first 5 epochs from 0 to 1.6, followed by a 10x drops of the learning rate at 30, 60 and 80 epochs. For LARS, we use warmup learning rate over 20 epochs for 4K and 16K batch sizes, 25 epochs for 32K batch size with a polynomial decay (p=2) until end of training. For Shampoo we use the same layer-wise heuristics and hyperparameters as LARS with Grafting such that the direction is changed to the one computed by Shampoo. We make use weight decay with value: $\lambda_2 = 2 \times 10^{-4}$ and label smoothing of $10^{-1}$.

## G.5 DETAILED RESULTS FOR EXPERIMENTS

Approximate wall clock times for the various tasks are as follows:

| Task | Model | Baseline | Shampoo |
|---|---|---|---|
| Recommendations: Criteo-1Tb | DLRM | 13 min | 8.2 min |
| Translation: WMT-14 En-Fr | Transformer | $\approx$ 12 hrs | 6.5 hrs |
| Translation: WMT-14 En-Fr | Transfomer-Big | $\approx$ 47 hrs | 29.5 hrs |
| Language Modeling: Wikipedia+Books | BERT-Large | 228 mins | 219 mins |

## G.6 BREAKDOWN OF STEP-TIME IN FIG. 2B

Each step of training consists of the following phases, whose times are shown in Fig. 2b.

- Forward Pass: Each core independently computes the predictions for each training example in its sub-batch.

- Gradient: The gradient is for the sub-batch is computed using the back-propagation algorithm.

- All reduction: The gradients for the sub-batches from all cores are averaged to compute the gradient for the minibatch. This is then sent back to each core.

- Preconditioner statistics: The preconditioner statistics for adaptive algorithms are updated, e.g. for AdaGrad, we set $H_i := H_i + g_i^2$ for all parameters, while for Shampoo, we set $L_i := L_i + GG^\mathsf{T}$ etc.

- Preconditioned gradient: The preconditioned gradient is computed - e.g. for AdaGrad, we compute $g_i/\sqrt{H_i}$, while for Shampoo, we compute $L^{-1/4}GR^{-1/4}$.

- Parameter updates: The parameters are updated using the preconditioned gradients. This step is the same for all algorithms: $W := W - \eta \tilde{G}$, where $\tilde{G}$ is the preconditioned gradient.

Note that the Shampoo computation of the preconditioners $L^{-1/4}, R^{-1/4}$ is pipelined on the host CPU, so does not show up in the step times.

