# OpenReview forum: "Towards Practical Second Order Optimization for Deep Learning"
_ICLR.cc/2021/Conference — Reject_

### Official Review · AnonReviewer1 · 2020-10-28
**Official Blind Review #1**

**Rating:** 7
**Confidence:** 2

**Review:**

The paper tackles the important problem of making second-order methods practical for stochastic optimization. The main contributions of the paper include an improvement to a previously proposed Shampoo algorithm that is, on a high level, an extension of AdaGrad to include a second matrix moment (such as covariance of gradients).

The paper addresses challenges along three axes. On the algorithmic level, computing the preconditioners is expensive both in terms of memory and compute and the authors suggest improvements theoretically motivated techniques to make this faster. On a numerical level, the authors put forth a case for investing in 64-bit arithmetic in hardware implementations required for computing the p-th root. This is an important point to make considering that the field has been trending towards the opposite direction and risks a local maxima. On the distributed implementation level, the authors implement the preconditioning computation (done every few steps) asynchronously on the host CPU owing to lack of double precision support on the accelerators attached to the host (GPU/TPU).

Experiments are run on a varied set of tasks (translation, CTR, NLP, vision) and they show fairly significant reduction in the number of optimization steps. Overall, the paper tackles the important problem of second order optimization in a more practical way. The motivations, main contribution and supporting experiments are well laid out.

---

> ### Author Response · Authors · 2020-11-11
> **Thank you for your review!**
>
> We also find that the field is moving towards a local maxima with respect to the techniques being explored, and hence we showed results in improving steps to convergence on multiple domains from the standard MLPerf benchmarks. We are seeing walltime improvements on several of these, and more will follow as hardware and software starts catching up.  Moreover, as you pointed out, our experiments show that our method generalizes to multiple domains with standard benchmarks -- before this each domain had its own SOTA optimizer. We would like to know if we could provide any more information to improve the confidence in the rating.

---

### Official Review · AnonReviewer3 · 2020-10-28
**An interesting approach for making second order methods practical, but needs more clarification**

**Rating:** 7
**Confidence:** 4

**Review:**

######################################################################

1.  Paper Summary

This work addresses practical challenges in applying full matrix pre-conditioner methods (such as Shampoo) on problems involving large datasets and architectures trained using a distributed setup.  In particular, this work presents a practical extension for the Shampoo algorithm by (1) using only a left or right preconditioner for large layers (2) computing inverse pth roots via coupled Newton iteration algorithms (3) distributing preconditioner computation across CPU cores in a CPU-GPU/TPU cluster and (4) delaying preconditioner computation to occur only once per several steps.  The proposed modifications lead to an implementation of Shampoo that consistently decreases the number of training steps and in certain cases provides a direct wall time improvement over Adagrad/Adam.

######################################################################

2. Strengths

2.1. The proposed modifications do seem to make Shampoo practical when training large models on large datasets using a distributed setup, and the authors present evidence that their implementation does converge faster (in number of steps and occasionally wall time) than Adam and Adagrad.  The authors also present an example of CTR prediction where their method produces state of the art results.

2.2. A few of the modifications are mostly well founded: namely, distributing preconditioner computation across unused CPUs and using the coupled Newton iteration for computing the inverse pth root.

######################################################################


3. Limitations/Questions

3.1. Some of the modifications seem to be heuristic and I found it unclear why these should work well in practice.  In particular, while I found Lemma 1 an interesting theoretical result, it was unclear to me why the proposed preconditioner was actually a good approximation for the full preconditioned matrix from Adagrad.  Does this somehow follow directly from the regret bound provided in the appendix? It appears that these regret bounds only show that Shampoo with the provided extensions converges, but there could be still be a gap between the performance of the modified Shampoo and original Shampoo.  It would be helpful if the authors could comment on this.  I felt similarly about the delayed preconditioner computation: The fact that computing the preconditioner only once every several epochs is interesting, but I didn't quite follow why this should have worked in the first place.  Is there some further intuition the authors could provide for this phenomenon?

3.2. While the authors demonstrate that their modifications worked well on the settings they described on the bottom of page 2, it appears that there are a lot of elements to tune (described in the appendix) to ensure that their method works well in these settings.  It would be helpful if the authors could clarify a bit more about the basic elements of their method that require significant tuning.  For example, while I understand that there are several works using learning rate scheduling to achieve state of the art results, in most experiments, Adam can be used with a learning rate of 1e-4.  In the proposed method, how often (in number of steps) should one compute the preconditioner matrix? I understand that there is a plot comparing this in Figure 3c, but on smaller datasets, do we actually need to compute the preconditioner often in order to get convergence?

3.3. Point 3.2 leads to my next concern about the work.  Namely, if the preconditioner does need to be computed more frequently, this would lead to a significant increase in wall time on smaller datasets.  In particular, the authors even mention that there is in fact no wall time improvement on full ImageNet due to a lack of benefit of amortization (I'm assuming this is referring to the frequency of preconditioner computation prior to convergence, but please correct me if I'm mistaken).  Thus, the practicality of the proposed method seems limited to extremely large datasets, and may be of practical use only to large organizations with access to resources capable of handling such large datasets.  However, for several of these organizations, it may be easier to just parallelize a simpler method across more GPUs/TPUs.  Thus, I would be interested in hearing more of a discussion about concrete hardware/software improvements that would boost the performance of the current algorithm over currently used algorithms.  For example, is the main suggested improvement to provide better support for double precision computations? I feel that most other hardware improvements would benefit first order methods anyway.


######################################################################

4. Score and Rationale


Updates: I have changed my score in light of the author responses and edits and now am in favor of accepting this work.

Currently, I vote for rejecting.  However, my decision is borderline.  My main concerns with the work are (1) the practical relevance of the work as compared to first order methods and (2) the rationale around the heuristics proposed in the current method.  In particular, I am concerned that while the proposed method consistently improves the number of steps, it does not necessarily improve wall time computations even on ImageNet (which would be considered a large dataset by many organizations).  I am also unsure as to how software/hardware benefits could boost the performance of the current method without significantly boosting the performance of first order methods as well.  I am definitely open to changing my review provided that the authors are able to provide clarification around these and other points in section 3.


######################################################################


5. Minor Comments

5.1. Apart from the points in section 3, it would be helpful to have a discussion section at the end of the work.

5.2. I believe the y-axis label is missing in Figure 3 (though I believe it should just be -log(perplexity)).


######################################################################

Post-Rebuttal Updates:

6. Thank you for running the experiments on small datasets and for clarifying my additional concerns.  In light of your clarifications, I do find this work to be a practical extension of Shampoo and am in favor of accepting the work.  I am still surprised by the fact that delayed pre-conditioner computation still leads to improved convergence over 1st order methods, but the presented empirical evidence demonstrates a consistent benefit in wall time across a number of settings.

---

> ### Author Response · Authors · 2020-11-11
> **Thank you for your review!**
>
> 3.1 Lemma 1 is intended to show that the extension works in the convex setting. In the non-convex setting, we verify this empirically, on the embedding layers for DLRM, as well as Transformers. In fact, without modifications proposed in Lemma 1 and Lemma 2, the original Shampoo method would not be practical for these models.
>
> A bit of clarification here: We update the preconditioners many times within each epoch.  In particular, for Transformers-Big we update the preconditioners 24 times per epoch (since we update the preconditioners every 1000 steps and one epoch is ~24K steps). For criteo we update the preconditioners ~64 times per epoch (since we update the preconditioners every 999 steps and one epoch is ~64k steps).
>
> Why does it work to update the preconditioners infrequently? Our intuition is that the loss curvature does not change dramatically after each step. We show experimentally in Figure 3.c. that we are able to get similar quality results at every 1000 steps compared to running every 100 steps. However, this is a performance and quality tradeoff. The only way to increase the frequency of computing preconditioners is with better hardware/software support.
>
> 3.2. We will include tuning guidance in our revision. Note that hyperparameters for the baselines have been heavily tuned by several large organizations (https://github.com/mlperf/training_results_v0.7). The additional Shampoo hyperparameters only tradeoff quality with efficiency -- our guidance is to set them to the smallest value that results in competitive performance. The "frequency of computing preconditioners" is easy to set, as the number of training steps that can be completed in the amount of time needed to compute the largest inverse pth root. Hence, faster implementation via Hardware/Algorithm of pth root allows setting smaller values. For smaller models, pth root should run much faster, as complexity is cubic in the dimension of the statistics - hence we expect we can run it more often.
>
> 3.3 On improving Walltime on ImageNet which is considered a large dataset by many organizations:
>
> Smaller datasets: for smaller datasets with less hardware our method will produce  wall time improvements. In our Imagenet experiment, we mimicked the MLPerf standard benchmarking setup: parallelizing over 256 cores TPUs v3 at 32K batch size. We trained for 2262 steps (training only takes a few minutes), and needed to run p-th roots every 10 steps, hence TPUs needed to wait for the pth root computations to complete, and the data transfer cost was significant. On a smaller hardware configuration, this wait time would be eliminated and the data transfer would be a much smaller fraction of the overall compute time, hence Shampoo would see the wall-time improvements as well. Moreover, our suggested hardware/software improvements can help with wall-time improvements for small and large datasets.
>
> Easier to parallelize first order methods: Our method reduces total training costs dramatically -- reducing training time with fixed hardware.  One can improve the wall time by parallelizing simpler methods as you suggest, but only by dramatically increasing hardware costs.
>
> Examples of hardware/software improvements: We will include this in the discussion section of our paper.
>
> (a) Implementation of first order methods are highly optimized already in multiple frameworks. For example, weight update sharding pattern matches first order methods, (arxiv.org/abs/2004.13336) and dramatically reduces the time spent in the update step as well as memory used.  This change can also be applied to Shampoo with blocked preconditioners -- but we do not have support for it yet as it requires compiler level support, and is not expressible at the program layer. Currently every core must update all layers which is quite inefficient.
>
> (b) Exploiting symmetric matrices in the compiler. We haven't found support for typing operands as symmetric, which can reduce compute flops and storage by upto ~50%.  This is specific to second order methods.
>
> (c) Mixed precision algorithms may work for inverse pth roots and can help increase the frequency of preconditioner computation.
>
> Hardware:
> (a) Increased memory per chip can allow larger preconditioners.
>
> (b) Hardware support for Float64 can in fact allow for running inverse very often. Our estimate is if we had ~10 TFlops of f64 compute on chip: we can run 512x512 matrix inverse pth root in less than 1ms via the coupled newton method compared to 300ms we observe on CPUs (highly optimized, AVX2 etc) - not including the cost of transfer to CPU via PCI-E.
>
> (c) Hardware support for storing/packing and using upper/lower triangular matrices efficiently, as available in LAPACK.
>
> These optimizations have never been implemented on neural network accelerator hardware/software as first order methods do not need them. These help Shampoo become faster w.r.t. first order methods, which have already been optimized heavily.

---

> > ### Comment · AnonReviewer3 · 2020-11-15
> > **Thank you for the prompt response**
> >
> > Regarding the results of Lemma1, Theorem 3, and Theorem 4: Thank you for clarifying that the proof implies that the extension converges in the convex setting.  I feel that in light of this result and that the experiments are in the non-convex setting, the text in 3.1 should be clarified.  Namely, the current text in Section 3.1 on page 4 makes it seem as though lemma 1 provides an approximation for the Shampoo preconditioners in all settings.  I feel that it should be made explicit that the convergence results are in the convex setting and clarify that this serves as an intuition for why this approximation may be useful.
> >
> > Regarding preconditioner computation and concern 3.2:  I feel that the discussion provided in your comment above is important and would be useful to clarify further in the text (for example in the delayed preconditioners section on page 5).
> >
> > Regarding smaller datasets:  Would you be able to provide some concrete examples of the wall time improvement on a smaller datasets?  For example, training simplenet or resnets on CIFAR10/CIFAR100 or even training resents on ImageNet using far fewer TPUs/GPUs (like a system with 1 or 2 GPUs but with many CPU cores).  I think this would likely strengthen the message significantly.  I am trying to understand whether the proposed modifications would make Shampoo a viable method for a more general audience (outside of organizations with large distributed systems setups).  Is the current limitation the number of CPU cores relative to the number of TPUs/GPUs available to you?
> >
> > Regarding examples of hardware/software improvements: I found your concrete examples above extremely helpful, and I feel that they should definitely be added to at least the supplementary.

---

> > > ### Author Response · Authors · 2020-11-18
> > > **Thanks again for your response!**
> > >
> > > We have addressed the comments you had and have updated the paper.
> > >
> > > We have clarified that the Lemmas are for the convex setting as well as the intuition on how to pick the frequency of the preconditioner, and have incorporated the discussion into Page 5.
> > >
> > > Results on smaller datasets continue to show that our modifications work in practice:
> > >
> > > We have added results on CIFAR-10 with ResNet-50 on 2 cores of CloudTPU-v2 at batch size 2048. We see an overall training time reduction of 42% (1428 seconds to 827 seconds) - Note that the longest time taken for preconditioner inverse computation is less than 1ms on the CPU. We use a total of 8 CPU cores to run the inverses.
> > >
> > > Is the number of CPUs a bottleneck?
> > > We are not bound by the number of CPUs but rather by the performance of each CPU core, as there is limited parallelism we can exploit for each of the operations in the inverse pth root, or in singular value decomposition algorithms. Note that mixed precision and algorithmic improvements can speed up the latency of computing the preconditioner.
> > >
> > > We have updated our concluding remarks to include the concrete list of hardware and software improvements for second order methods.

---

### Official Review · AnonReviewer4 · 2020-10-28
**The contributions of this paper are not so clear and appear to be incremental if only an iterative perconditioning method is proposed.**

**Rating:** 6
**Confidence:** 3

**Review:**

Main idea: The paper developed a practical second-order preconditioned method, which is based on the Shampoo algorithm, to improve the wall-clock time compared with state-of-the-art first-order methods for training deep networks.


1.	This paper is heavily drawn from the shampoo algorithm, algorithm designing, and even the later theoretical analysis (such as theorem 3). It appears that the iterative preconditioning method is only incremental improvement over the Shampoo algorithm. However, the iterative method is widely used in the similar spirit of the limited memory of BFGS or LU decomposition.
2.	The main text of this paper has only two lemmas. What is the main theorem of this paper?
3.	Even though the paper mentioned wall-time everywhere, all the improvements shown in the figures, compared with Adagrad and Adam, are in terms of steps, such as Figure 2(a), Figure 3 - 5 and Table 1. The authors should directly show wall-clock time.
4.	Some minor comments: fonts in Figure 1 are too small; in other figures, the fonts in x-axis and y-axis are also very small; in figure 2 (b), the author should explain more about how to break down the latency of a single step into the parts mentioned in the figure.

##################################

Updates: I have changed my score after reading the author's responses.

---

> ### Author Response · Authors · 2020-11-11
> **Thank you for your review!**
>
> 1. We extend the Shampoo algorithm for large models in multiple domains (Recommendations, Image Classification, Translation, and Language modeling) and showcase improved steps to convergence, and walltime for some of them. We contribute the following extensions for handling large models. Without these extensions, it is not possible to fit the preconditioners in memory, nor would the inverse root computation finish in reasonable time.
>
>   (a) Blocked preconditioning for extremely large layers that occur in Transformer-Big models: results in Figure 3.b, and Lemma 2. Notice that blocking does not affect the quality for this model, as there is a tradeoff between rank (block size) and block count.
>
>   (b) Extension to large embedding layers (186M items): see Lemma 1 and Appendix D for efficient implementation. We show walltime improvement and produce a new SOTA result on the Criteo 1TB dataset -- we show 0.3% improvement in AUC on the standard DLRM model (Figure 5.a), whereas recent work by SambaNova Systems gets 0.2% improvement only by increasing the model size by 8x.
>   https://sambanova.ai/articles/surpassing-state-of-the-art-accuracy-in-recommendation-models/
>
>   Apart from these algorithmic extensions, we provide a systems contribution on how to effectively make use of heterogeneous computing (CPUs, and TPUs/GPUs), which allows us to make this second order method work well for multiple domains.
>
>   Another contribution is on guidance on where to move hardware and software design. On the hardware front, we provide evidence for the benefit of high precision arithmetic for ML, counter to the prevailing wisdom:
>   "It is well appreciated that in the presence of statistical approximation and estimation errors, high-precision computation in the context of learning is rather unnecessary (Bottou & Bousquet, 2007)"
>    --- http://proceedings.mlr.press/v37/gupta15.pdf
>   "While machine learning provides much of the impetus for the development of half precision arithmetic in hardware."
>    --- https://pdfs.semanticscholar.org/bf9d/4cf8ba3dd22a80e782dfd059e677df7c1629.pdf
>
>   The algorithmic changes to the coupled Newton iteration (Appendix E), such as warm starting and other changes to improve stability, do help with run time. However, it is possible to use other well known methods to compute inverse pth roots (like Schur-Newton).
>
> 2.  The main theorem, which follows immediately from the lemmas, is that the Shampoo converges even with these extensions. These extensions are important because the vanilla Shampoo algorithm from ICML'18 does not work on many models e.g. in that paper the authors only showed training loss on CIFAR10.
>
> 3. We will add a table on wall clock times to the paper in our revision. Here are the exact numbers.
>
>   | Task					                           | Baseline&nbsp;&nbsp; | Our improvement  |
>   |:-  | :-: | :-: |
>   | Recommendations: Criteo-1Tb with DLRM 	 	                    |  13 mins |  	8.2 mins |
>   | Translation: WMT-14 En-Fr with Transformer             |    ~12 hrs |  6.5 hrs  |
>   | Translation: WMT-14 En-Fr with Transfomer-Big.      |  ~47 hrs  | 29.5 hrs  |
>   | Language Modeling: Wikipedia and Books with BERT-Large &nbsp;&nbsp;  |  228 mins  | 219 mins |
>
> 4. We will fix this in our revision, and add a description of the latency breakdown in the appendix.

---

### Author Response · Authors · 2020-11-18
**Revised manuscript**

We have updated the paper to address comments from all reviewers. We would really appreciate if reviewers could send us any additional comments they may have.

Thank you all again for your reviews!

---

### Decision · Program_Chairs · 2021-01-07
**Final Decision**

**Decision:**

Reject

**Comment:**

Dear authors,

I like to topic of your paper very much. Indeed, your work is trying to show that 2nd order methods can be efficiently implemented in a distributed environment and can achieve improvement in training times.

However, having worked on distributed computing for many years, I personally think that reporting running time in your work is not very informative (without mentioning what hardware is used during computation), and one cannot understand the connection or reproduce your results. Also, it was not clear how the baselines were implemented, and how the hyper-parameters were tuned. It is also not clear why you haven't picked better benchmarks to compare your work.

I think that addressing both the concerns from reviewers and the one mentioned above would improve the paper significantly. I would really like to see if accepted in the near future after these issues are fully addressed.

---

> ### Author Response · Authors · 2021-01-15
> **Please read the paper.**
>
> Thank you for changing your words to be professional.
>
> You brought up four points that you used to override reviewer consensus, including reviewers who actively engaged with us during the author discussion. We made our first response on 11 Nov 2020 and completed our responses by 17 Nov 2020, which gave you about a week if you had these further concerns. Anyway, we directly address the points you made with quotes from the paper.
>
> 1. Your work is not very informative (without mentioning what hardware is used during computation), and one cannot understand the connection or reproduce your results.
>
> A description of the hardware setup for each experiment was included on Page 20, Table 3 including the number of TPU cores used in training for each one of our experiments.
>
> Page 6 contains further description for Transformer experiments: "The experiment was run on 32 cores of a Cloud TPU v3 Pod, and the implementation of the optimizer was carried out in the Lingvo (Shen et al., 2019) sequence to sequence modeling based on TensorFlow".
>
> For CIFAR-10, which was included at the request of reviewers, we mention on Pg 20: "We train a ResNet-50 model on CIFAR-10 (Krizhevsky et al., 2009) with 2 cores of CloudTPU-v2 at batch size 2048."  This was included for replicating small scale setup (for eg: practicality of our method on 1 GPUs).
>
> CloudTPU-v3 is available for public consumption. https://cloud.google.com/tpu/docs/tpus . Moreover, several papers are being published today in ICLR 2021 with CloudTPUs. We had mentioned on page 6 that we would open-source the code before publication (a version of the Shampoo code is already in TensorFlow Lingvo library in open source -- but to not break anonymity we chose to not mention it in the paper).
>
> ######################################################################
>
> Your next three questions are answered together:
>
> 2. Also, it was not clear how the baselines were implemented,
> 3. how the hyper-parameters were tuned
> 4. haven't picked better benchmarks to compare your work.
>
> We used industry standard baselines in 3 cases, and mentioned that the code is from MLPerf v0.7 (July 2020).
>
> ImageNet  Pg 2,  "Image classification: we achieve MLPerf target accuracy of 75.9% (Mattson et al., 2019) at 32K batch size on the standard ResNet-50 ImageNet benchmark in 10% fewer steps than previous state-of-the-art."
>
> Criteo: Pg 7,  "We compared Shampoo against the highly tuned SOTA baseline from MLPerf v0.7 training benchmarks (Wu et al., 2020)."
>
> BERT-Large Pg 5, "BERT-Large, DRLM, as well as ResNet-50 used the MLPerf v0.7 Tensorflow baselines (Mattson et al., 2019)"
>
> All the code for the baselines are publicly available, we mention MLPerf v0.7 several times! (https://github.com/mlperf/training_results_v0.7/ and for eg: ResNet: https://github.com/mlperf/training_results_v0.7/tree/master/Google/benchmarks/resnet/implementations/resnet-research-TF-tpu-v4-128).
>
> These baselines that we beat with Shampoo are highly tuned -- we mention this also. These achieve the current known best timings as well as steps to convergence, and their hyperparameters have been tuned by several teams from many companies (and the hyper-parameters are also publically available).  Our hyperparameters are mentioned in Page 20, Table 3.
>
> For the Transformer on WMT'14 experiments we cite the paper we took the baselines from. "We used the state-of-the-art Transformer architecture (Vaswani et al., 2017). This architecture contains 93.3M parameters and consists of 6 layers for its encoder and decoder. Each layer is composed of 512 model dimensions, 2048 hidden dimensions, and 8 attention heads."
>
> Regarding tuning this model, we mention on Pg 19 "For the smaller Transformer experiments, we tuned the hyperparameters for each algorithm over 100 trials. We took the best settings for the momentum and second-moment parameters, and tuned the learning rates until either the model became unstable, or did not increase performance. For Shampoo, we used a per layer learning rate derived from AdaGrad (see Appendix G for details), and found that for the exact same hyperparameter settings as AdaGrad, Shampoo provides a modest improvement in performance. Moreover, Shampoo allows for larger learning rates than AdaGrad does, as shown in Fig. 4a".